# Hypocalcemia and Vitamin D Deficiency amongst Migraine Patients: A Nationwide Retrospective Study

**DOI:** 10.3390/medicina55080407

**Published:** 2019-07-25

**Authors:** Urvish Patel, Nishanth Kodumuri, Preeti Malik, Amita Kapoor, Princy Malhi, Kulin Patel, Saleha Saiyed, Liseth Lavado, Vinod Kapoor

**Affiliations:** 1Department of Neurology & Public Health, Icahn School of Medicine at Mount Sinai, New York, NY 10029, USA; 2Department of Neurology, University of South Carolina School of Medicine, Columbia, SC 29209, USA; 3Department of Public Health, Icahn School of Medicine at Mount Sinai, New York, NY 10029, USA; 4Department of Neurology, Bayonne Medical Center CarePoint Health, Bayonne, NJ 07002, USA; 5Department of Nursing, Holy Family University, Newtown, PA 18940, USA; 6Department of Nursing, Rutgers University School of Nursing, Newark, NJ 07107, USA

**Keywords:** migraine, hypocalcemia, vitamin D deficiency, nationwide inpatient sample

## Abstract

*Background and Objectives*: Inadequate vitamin D and calcium intake have been linked to many health issues including chronic headaches. Some studies suggested an association between low vitamin D levels and increase the risk of frequent headaches in middle-aged and older men; however, no single study reported the role of these deficiencies in migraine patients. We aimed to investigate the association of hypocalcemia and vitamin D deficiency with migraine hospitalizations. *Materials and Methods*: A population-based retrospective cross-sectional analysis of the Nationwide Inpatient Sample (NIS) (years 2003–2014) in migraine hospitalizations was performed. The prevalence, demographic characteristics and All Patient Refined Diagnosis Related Groups severity/disability association were compared in patients with hypocalcemia and vitamin D deficiency to those without deficiencies, using ICD-9-CM codes. Weighted analyses using Chi-Square, paired Student’s *t*-test, and Cochran–Armitage trend test were performed. Survey logistic regression was performed to find an association between deficiencies and migraine hospitalizations and deficiency induced disability amongst migraineurs. *Results*: Between years 2003 and 2014, of the total 446,446 migraine hospitalizations, 1226 (0.27%) and 2582 (0.58%) presented with hypocalcemia and vitamin D deficiency, respectively. In multivariable analysis, hypocalcemia [Odds Ratio (OR): 6.19; Confidence Interval (CI): 4.40–8.70; *p* < 0.0001] and vitamin D deficiency (OR: 3.12; CI: 2.38–4.08; *p* < 0.0001) were associated with markedly elevated odds of major/extreme loss of function. There was higher prevalence (3.0% vs. 1.5% vs. 1.6%; *p* < 0.0001) and higher odds of migraine among vitamin D deficiency (OR: 1.97; CI: 1.89–2.05; *p* < 0.0001) patients in comparison to patients with hypocalcemia (OR: 1.11; CI: 1.03–1.20; *p* = 0.0061) and no-deficiency, respectively. *Conclusions*: In this study, we demonstrated a significant association between hypocalcemia and vitamin D deficiency with migraine attacks and deficiency induced loss of function among migraineurs. Early preventive measures may reduce the disability in migraineurs.

## 1. Introduction

Migraine is a disabling headache disorder characterized by unilateral pulsating headache associated with photophobia, phonophobia, nausea and occasionally transient neurological symptoms [1]. Globally, recurrent migraines affect 6% of men and 18% of women [2]. In addition, one out of four adults in the United States suffer from intense headaches [3]. Migraine has significant impact on patient’s quality of life, thereby affecting work and family life [4]. There are multiple triggers for migraine which include environmental and genetic factors, but pathophysiology of migraine is not yet well established [5]. However, there are few proposed causes of migraine such as Methylene Tetrahydrofolate reductase (MTHFR) gene mutation, low serotonin level, increased calcitonin gene related peptide (CGRP) and low vitamin D levels [2]. The migraine and calcium metabolism association was first described in 1934 [6]; since then, many studies have strengthened the association of migraine and calcium metabolism/vitamin D [7]. Sohn et al. published a similar study reporting vitamin D deficiency is common in cluster headache but exact role of deficiency was uncertain [8]. It is believed that inflammation is involved in the pathogenesis of both migraine and vitamin D deficiency [7], which is further strengthened by the inverse association between C-reactive protein and vitamin D levels [9]. Furthermore, vitamin D and calcium supplementation has been associated with better outcomes in terms of number and intensity of migraine [10,11]. On the contrary, a recent review mentioned that there is no scientific evidence of vitamin D levels and migraine association [12].

In this retrospective cross-sectional study, we hypothesized that migraine is associated with low vitamin D and Calcium levels. The primary objective of this study was to investigate if low vitamin D and Calcium levels predict disability in the subset of large inpatient sample with 446,446 migraine hospitalizations. The secondary objective was to study whether these deficiencies had any association to worsen migraine which led to migraine hospitalizations.

## 2. Materials and Methods

Data were obtained from the Agency for Healthcare Research and Quality’s Healthcare Cost and Utilization Project (HCUP) NIS files between January 2003 and December 2014. The NIS is the largest publicly available all-payer inpatient care database in the United States and contains discharge-level data provided by states that participate in the HCUP (including a total of 46 in 2011). This administrative dataset contains data on approximately 8 million hospitalizations in 1000 hospitals that were chosen to approximate a 20% stratified sample of all US community hospitals, representing more than 95% of the national population. Criteria used for stratified sampling of hospitals into the NIS include hospital ownership, patient volume, teaching status, urban or rural location, and geographic region. Discharge weights are provided for each patient discharge record, which allow extrapolation to obtain national estimates. Each hospitalization is treated as an individual entry in the database and is coded with one principal diagnosis, up to 24 secondary diagnoses, and 15 procedural diagnoses associated with that stay. Detailed information on NIS is available at http://www.hcup-us.ahrq.gov/db/nation/nis/nisdde.jsp.

### 2.1. Study Population

We used the 9th revision of the International Classification of Diseases, clinical modification code (ICD-9-CM) to identify adult patients admitted to hospital with a primary diagnosis of migraine (ICD-9-CM code 346). Similarly, patients with hypocalcemia and vitamin D deficiency were identified as secondary diagnosis associated with migraine and not as a co-morbidity using ICD-9-CM codes 275.41 and 268, respectively. We used ICD-9-CM codes to identify independent predictors (covariates), including the comorbidities of hypertension, diabetes mellitus, hypercholesterolemia/lipidemia, chronic use of NSAIDs, smoker (current/past), drug abuse, AIDS and alcohol abuse/dependent. Appendix A lists all ICD-9-CM codes that were used for this study. Age < 18 years and admissions with missing data for age, gender, and race were excluded. The sample size was based on the available data.

The data were taken from the Nationwide Inpatient Sample, which is a deidentified database from “Health Care Utilization Project (HCUP)” sponsored by the Agency for Healthcare Research and Quality, USA, thus informed consent or IRB approval was not needed for the study. The relevant ethical oversight and HCUP Data Use Agreement (HCUP-4Q28K90CU) were obtained for the study.

### 2.2. Patient and Hospital Characteristics

Patient characteristics of interest were gender, age, race, insurance status and concomitant diagnoses as defined above. Race was defined by white (referent), African American, Hispanic, Asian or Pacific Islander, and Native American. Insurance status was defined by Medicare (referent), Medicaid, Private Insurance, and Other/Self-pay/No charge. We defined the severity of co-morbid conditions using Deyo’s modification of the Charlson co-morbidity index (Appendix A). Facilities were considered to be teaching hospitals if they have an American Medical Association–approved residency program, are a member of the Council of Teaching Hospitals, or have a full-time equivalent interns and residents to patient’s ratio of ≥0.25. HCUP NIS contains data on total charges for each hospital in the databases, which represents the amount that hospitals billed for services.

### 2.3. Outcomes

We tried to find the loss of function (LoF), length of stay (LoS), and cost of hospitalization associated with these deficiencies amongst migraine hospitalizations (years 2003–2014). The comparison of disability/loss of function was investigated by All Patient Refined Diagnosis Related Groups (APR-DRGs) severity between patients with hypocalcemia (Ca) and vitamin D deficiency (vitD) and patients without deficiencies. APR-DRGs were assigned using software developed by 3M Health Information Systems, where score 1 indicates minor loss of function; 2, moderate; 3, major; and 4, extreme loss of function.

Our secondary outcome of interest was to establish link between migraine and deficiencies (hypocalcemia and vitamin D deficiency) amongst January 2014–December 2014 hospitalizations. To evaluate odds ratio of relation whether deficiencies leads to migraine hospitalization, patients with and without migraine were evaluated for deficiencies. The reason to choose only year 2014 data was the large number of US hospitalizations (more than 20 million) each year with and without migraine and inability of software to run analysis for 12 years of data of total US admissions.

### 2.4. Statistical Analysis

All statistical analyses were performed using the weighted survey methods in SAS (version 9.4). Weighted values of patient-level observations were generated to produce a nationally representative estimate of the entire US population of hospitalized patients. *p*-values of <0.05 was considered significant. Univariate analysis of differences between categorical variables was tested using the chi-square test and analysis of differences between continuous variables (LoS and Cost of hospitalization) was tested using paired Student’s *t*-test. Mixed-effects survey logistic regression models with weighted analysis were used for the categorical dependent variables, including migraine and outcomes of interest, in order to estimate odds ratio (OR) and 95% Confidence Interval for the association between deficiencies and migraine in year 2014 cohort as well as deficiencies and deficiency induced loss of function amongst migraine hospitalizations during years 2003–2014.

The hierarchical models (demographics and patient-level factors nested within hospital-level factors) were created as random effects with in the model for the outcomes. We included demographics (age, gender, and race), patient-level hospitalization variables (admission day, primary payer, admission type, and median household income category), hospital-level variables (hospital region, teaching versus nonteaching hospital, and hospital bed size), comorbidities (mentioned in Table 1), and concurrent conditions such as hypertension, diabetes mellitus, hypercholesterolemia, obesity, smoking status, drug abuse, alcohol abuse, long term use of NSAIDs, and Charlson’s co-morbidity index (CCI).

We investigated the link between migraine attacks and deficiencies and deficiencies induced loss of function by creating separate mix effect survey logistic regression models with weights to account for sampling strategy:

Table 3 shows the relationship of vitamin D deficiency and hypocalcemia with post-migraine major/severe loss of function amongst migraine hospitalizations for years 2003–2014 (Table 3A presents the unadjusted analysis and Table 3B presents the adjusted multivariable analysis).

Table 4 shows the relationship of migraine attacks with vitamin D deficiency and hypocalcemia amongst year-2014 hospitalizations (Table 4A presents the univariate analysis and Table 4B presents the adjusted multivariable analysis).

For each model, the area under the ROC curve/C-index (the area under ROC curve is a measure of goodness of fit for binary outcomes in a logistic regression model) was calculated. Higher values of AUC of ROC curves denote a better discriminating ability of a test or model between a true positive and a false positive value. All statistical tests used were two-sided, and *p* < 0.05 was deemed statistically significant. No statistical power calculation was conducted prior to the study.

## 3. Results

### 3.1. Analysis of Deficiencies Induced Loss of Function amongst Migraine Hospitalization (Year 2003–2014)

#### 3.1.1. Disease Hospitalizations

We found 446,446 hospitalizations (weighted) due to migraine during 2003–2014 after excluding patients with age < 18 years and admissions with missing data for age, gender, and race (Figure 1A). Out of 446,446 migraine hospitalizations, 1226 (0.27%) had hypocalcemia and 2567 (0.58%) had vitamin D deficiency.

#### 3.1.2. Prevalence Trends

We analyzed trends of hypocalcemia and vitamin D deficiency in migraine hospitalizations. As shown in Figure 2, trends of hypocalcemia were slightly increasing (hypocalcemia: 0.21% in 2003 to 0.32% in 2014; P-_Trend_ < 0.0001). However, trend of vitamin D deficiency has increased tremendously from 2003 to 2014 (vitamin D deficiency: 0.02% in 2003 to 1.59% in 2014; P-_Trend_ < 0.0001).

#### 3.1.3. Demographics, Patient and Hospital Characteristics, and Comorbidities

Migraine hospitalizations with hypocalcemia were more likely to be female (86.45% vs. 80.46%, *p* < 0.0001), white (76.44% vs. 73.69%, *p* < 0.0001), and Native American (1.27% vs. 0.52%, *p* < 0.0001) than those without deficiency. Prevalence of hypocalcemia was almost equally distributed among all median household income categories and not with only low-income category. Co-morbidities such as obesity (16.14% vs. 10.07%, *p* < 0.0001) and renal failure (4.96% vs. 1.92%, *p* < 0.0001) were higher in patients with hypocalcemia than those without deficiency. Most of the hospitalizations with hypocalcemia had CCI (Deyo’s Charlson Co-morbidity Index) of either 0 or 1. Urban non-teaching hospitals and the West region had higher prevalence of hypocalcemia amongst migraine hospitalizations. (Table 1).

Migraine hospitalizations with vitamin D deficiency were more likely to be male (88.52% vs. 80.46%, *p* < 0.0001), African American (17.66% vs. 14.54%, *p* < 0.0001), Native American (1.42% vs. 0.52%, *p* < 0.0001) and have Medicare (24.8% vs. 17.99%, *p* < 0.0001) than those without deficiency. Similar to hypocalcemia, prevalence of vitamin D deficiency was almost equally distributed among all median household income categories and not with only low-income category. Co-morbidities such as diabetes (18.19% vs. 12.84%, *p* < 0.0001), obesity (20.48% vs. 10.7%, *p* < 0.0001), hypertension (45.67% vs. 36.45%, *p* < 0.0001), and depression (28.12% vs. 20.18%, *p* <0.0001) were higher in patients with vitamin D deficiency than those without deficiency. Most of the hospitalizations with vitamin D deficiency also had CCI (Deyo’s Charlson Co-morbidity) of either 0 or 1. Migraine in the large, urban teaching hospital and in the Mid-West region were more likely to have vitamin D deficiency (Table 1).

#### 3.1.4. The Outcomes

Table 2 shows the outcomes of hypocalcemia and vitamin D deficiency among migraine hospitalizations. Outcomes were disability/loss of function, cost, and length of stay (LOS).

Overall, 49.58% and 43.86% of migraine hospitalizations had minor and moderate loss of function, respectively. The prevalence of moderate, major, and severe disability was higher among hypocalcemia and vitamin D deficiency patients. The overall prevalence of major/severe loss of function was 27.35% and 20.9% in hypocalcemia and vitamin D deficiency patients, respectively, in comparison to non-deficient patients with 6.43% (*p* < 0.0001).

Mean length of stay (3.6 days and 3.4 days vs. 2.8 days, *p* < 0.001) and total cost of hospitalization were higher ($22,590 and $24,913 vs. $18,185; *p* < 0.001) in hypocalcemia and vitamin D deficiency patients, respectively (Table 2).

#### 3.1.5. Regression Model Derivation

As shown in Table 3A, the unadjusted odds (OR) for major/severe loss of function or disability due to vitamin D deficiency among migraine were 3.79 (95%CI: 3.05–4.72; *p* < 0.0001) (Model 1) and due to hypocalcemia were 5.53 (95%CI: 4.14–7.37; *p* < 0.0001) (Model 2). After adjusting for basic demographic with patient-level variables, comorbidities, CCI, concurrent conditions, and hospital-level variables such as hospital region, teaching status, and bed size, hypocalcemia (OR: 6.19; 95%CI: 4.40–8.70; *p* < 0.0001) and vitamin D deficiency (OR: 3.12; 95%CI: 2.38–4.08; *p* < 0.0001) were associated with higher odds of major/severe disability compared to without deficiency (Table 3B).

Table 3B also lists multivariable analysis of predictors of disability in migraine hospitalizations with hypocalcemia and vitamin D deficiency. Weekend admissions (compared to weekday) (adjusted OR: 1.12; 95%CI: 1.04–1.21; *p* = 0.003), co-morbidities such as obesity (adjusted OR: 2.47; 95%CI: 1.64–3.73; *p* < 0.0701), peripheral vascular disease (adjusted OR: 1.79; 95%CI: 1.46–2.19; *p* < 0.0001), hypertension (adjusted OR: 1.79; 95%CI: 1.46–2.19; *p* < 0.0001), renal failure (adjusted OR: 2.53; 95%CI: 2.15–2.97; *p* < 0.0001), coagulopathy (adjusted OR: 13.66; 95%CI: 11.86–15.73; *p* < 0.0001), acquired immune deficiency syndrome (adjusted OR: 2.21; 95%CI: 1.59–3.09; *p* = 0.0124), and congestive heart failure (adjusted OR:6.08; 95%CI: 5.23–7.06; *p* < 0.0001) were significant predictors of disability in patients with deficiency in comparison to without, amongst migraine hospitalizations. Concurrent conditions/secondary diagnosis such as alcohol abuse/dependence (adjusted OR: 1.38; 95%CI: 1.07–1.77; *p* = 0.0132) and epilepsy (adjusted OR: 1.98, 95%CI: 1.78–2.20, *p* < 0.0001 were also significant predictors of disability amongst.

The AUC or C statistic of the ROC was used to validate the accuracy of the regressions. The AUC was 0.507, 0.505, and 0.819 for unadjusted Model 1, unadjusted Model 2 and adjusted Model 3, respectively. Adjusted Model 3B has c-index > 0.7, which indicates a good model.

### 3.2. Analysis of Linkage between Deficiencies and Migraine Attacks amongst Year 2014 Hospitalizations

#### 3.2.1. Disease Hospitalizations

We identified 28,212,820 total hospitalizations in year 2014 (Figure 1B), of which 242,515 (0.86%) had hypocalcemia, 418,970 (1.49%) had vitamin D deficiency, and 461,755 (1.64%) had migraine. Out of 461,755 patients with migraine, 3630 (0.79%) and 12,570 (2.7%) had hypocalcemia and vitamin D deficiency, respectively. Patients with vitamin D deficiency had higher prevalence of having migraine (3.0% vs. 1.5% vs. 1.6%; *p* < 0.0001) compared to patients with hypocalcemia and no-deficiency (Table 4A).

#### 3.2.2. Regression Model Derivation for Relationship of Deficiencies with Migraine amongst Year-2014 Hospitalizations

Table 4B shows the odds of having migraine attacks amongst patients with hypocalcemia, vitamin D deficiency and no-deficiency after adjusting patients’ demographics, patient and hospital characteristics, co-morbidities, and CCI. In regression analysis, odds of having migraine attacks were higher among vitamin D deficiency (OR: 1.97%; CI: 1.89–2.05; *p* < 0.0001) and hypocalcemia (OR: 1.11; CI: 1.03–1.20; *p* = 0.0061) patients in comparison to no deficiency.

Beside deficiencies, patients with drug abuse (aOR: 1.54; CI: 1.50–1.58; *p* < 0.0001), obesity (aOR: 1.42; CI: 1.40–1.45; *p* < 0.0001), hypertension (aOR: 1.35; CI: 1.32–1.38; *p* < 0.0001), AIDS (aOR: 1.36; CI: 1.19–1.56; *p* < 0.0001), hypercholesterolemia/lipidemia (aOR: 1.33; CI: 1.29–1.37; *p* < 0.0001), and current or past smoker (aOR: 1.33; CI: 1.31–1.35; *p* < 0.0001) had higher odds of having migraine attacks.

The AUC or C statistic of the ROC was used to validate the accuracy of the regressions. The AUC was 0.725, which indicates a good model.

## 4. Discussion

Migraine is a common neurological presentation in adolescents and young adults. It is reported to be the first leading cause of disability in adults < 50 years old [13]. Migraine is reported to be more prevalent in females due to hormonal influence [14]. According to a recent review, migraine has prevalence of 2.6–21.7%; however, there was a significant difference in the study groups [15]. The pathophysiology of migraine is not yet well established but several mechanisms are proposed to explain the onset of cortical spreading depression [16]. In 1934, the association of calcium metabolism and migraine was first described, where similarities between migraine pathophysiology and hypocalcemia (Tetany) and excitability of nervous tissues were mentioned [6]. Since then, many studies have strengthened the observation of calcium metabolism in migraine [7,10]. It is also reported that vitamin D deficiency and hypocalcemia are associated with migraine attacks [7].

In this study cohort, we demonstrated that migraine severity and disability is significantly associated with vitamin D deficiency and hypocalcemia even after adjusting for risk factors. These results further strengthen the above-mentioned vitamin D deficiency and hypocalcemia association with migraine attacks. The prevalence of vitamin D deficiency has increased from 2003 to 2014 and a similar trend was observed in the migraine hospitalizations. The prevalence of moderate, major, and severe disability was higher among hypocalcemia and vitamin D deficiency patients. According to previous studies, adequate vitamin D and calcium levels prevent cardiovascular diseases, hypertension and diabetes and low vitamin D and calcium levels are associated with increased morbidity [17]. Our results are in line with previous observational studies and case reports [7,10,11].

In this cohort, we found that the length of migraine hospitalization and cost of hospitalizations were more in migraineurs with low vitamin D and calcium levels.

It is also reported that vitamin D deficiency and calcium metabolism are associated with non-migraine headache [18,19], which could be due to dysregulation of the calcium metabolism leading to neuronal excitation. Neurogenic inflammation is thought to be involved with the migraine cascade of events [7,20,21] and the literature suggests vitamin D has a role in inflammation [16,22,23]. There could be neurogenic inflammation augmented by vitamin D deficiency, leading to increased severity of the migraine attacks. Furthermore, the observed phenomenon of neuronal hyperexcitability in hypocalcemia [24] could be contributing to the sensory symptoms in migraine and thus increasing the migraine attacks severity. Even though there is no proposed pathophysiological mechanism showing vitamin D deficiency and hypocalcemia contribution to initiation of migraine attack, these deficiencies could potentially contribute to the severity and recurrence of the attacks. Hence, patients might benefit from vitamin D and calcium supplementations. However, the benefit of supplementation is controversial [12], while there is a therapeutic benefit of magnesium supplementation, which is believed to be involved with calcium metabolism [2]. There is a clear involvement of calcium homeostasis and vitamin D in migraine pathophysiology but no evidence of supplementation benefits. Our results warrant a large randomized study to evaluate the therapeutic benefit over a period of time.

### Strengths and Limitations

To the best of our knowledge, this is the first and large population-based cross-sectional study to report the association of migraine severity with hypocalcemia and vitamin D deficiency. NIS data represent the largest inpatient database and our study reflects the trends of migraine hospitalizations across the nation and not confined to any geographical region or specific populations. Despite being a large study with good statistical power, this study has limitations. Data from clinical registries or administrative databases are obtained retrospectively by chart abstractions based on the discharge diagnosis codes, billing codes, etc., and hence susceptible to coding errors. Unlike other coding systems, APR-DRG coding system used in this study to assess the severity of illness is externally validated. It is a reliable method with accurate and consistent results and is widely used by hospitals, consumers, payers and regulators [25,26]. The diagnosis of vitamin D deficiency and hypocalcemia in the NIS database are physician documented with clinical evidence. However, it is not known if the hypocalcemia diagnosis is based on total calcium level or ionized calcium during the admission. This is one of the limitations of the study. Furthermore, length of hospital stays reported for migraine admissions could be related to other complications during the admission and, unfortunately, we do not have the exact duration of headache in these patients. However, most of these patients admitted to the hospital are younger than 50 years old and the prevalence of chronic disease that would complicate hospital stay such as cardiovascular diseases, renal disorders, and pulmonary disorders are less in this subset of the population.

## 5. Conclusions

These results indicate that the severity of migraine attacks and cost of acute treatments is more in migraineurs with vitamin D deficiency and hypocalcemia compared to migraineurs without deficiency. Thus far, there is limited evidence of benefit from supplementation of vitamin D and calcium in migraineurs. These results emphasize the need for a randomized controlled study to assess the benefit of supplementation in migraine attacks.

## Figures and Tables

**Figure 1 medicina-55-00407-f001:**
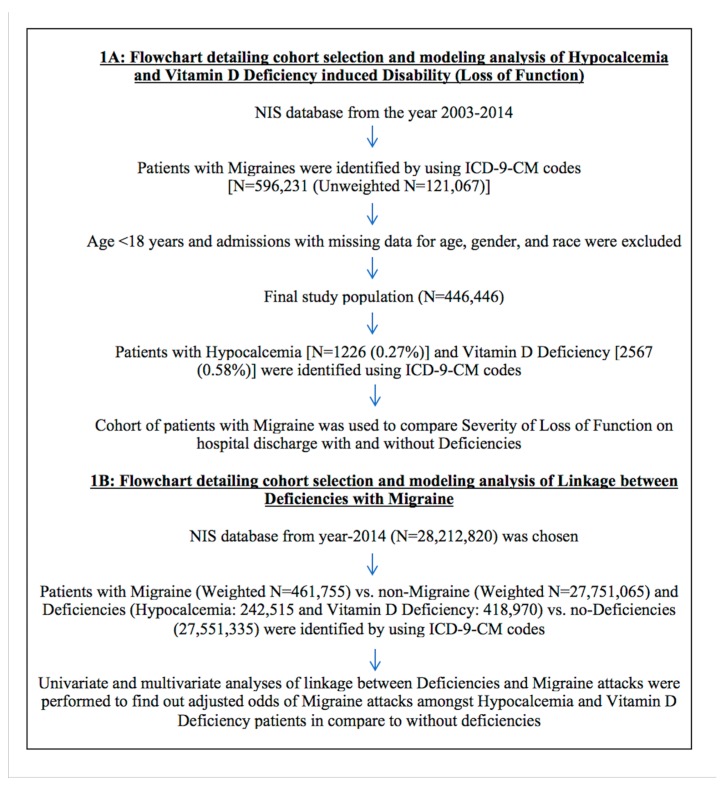
Flowchart detailing cohort selection and analysis modeling.

**Figure 2 medicina-55-00407-f002:**
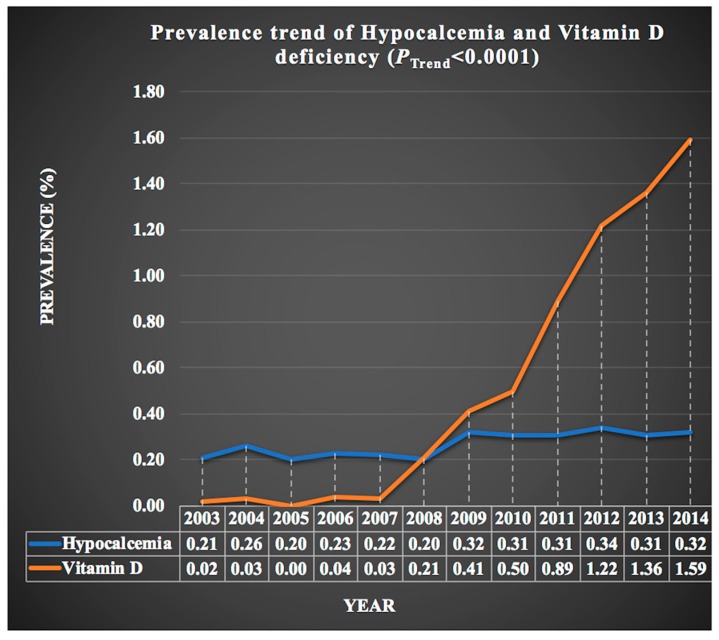
Prevalence trend of hypocalcemia and vitamin D deficiency from 2003 to 2014.

**Table 1 medicina-55-00407-t001:** Characteristics of vitamin D deficiency and hypocalcemia in migraine hospitalizations (years 2003–2014).

	Hypocalcemia	Vitamin D Deficiency	No Deficiency	Total	*p* Value
**Migraine (%)**	1226 (0.27)	2567 (0.58)	442,653 (99.15)	446,446	<0.0001
**Demographics of Patients**					
**Age Group (Years)**					<0.0001
18–34	363 (29.61)	493 (19.2)	116,523 (26.32)	117,379 (26.29)	
35–49	550 (44.86)	1029 (40.07)	182,478 (41.22)	184,057 (41.23)	
50–64	253 (20.64)	797 (31.04)	106,633 (24.09)	107,683 (24.12)	
65–79	50 (4.05)	204 (7.95)	29,762 (6.72)	30,016 (6.72)	
≥80	<10	45 (1.74)	7255 (1.64)	7310 (1.64)	
**Gender (%)**					<0.0001
Male	166 (13.55)	295 (11.48)	86,482 (19.54)	86,943 (19.47)	
Female	1060 (86.45)	2272 (88.52)	356,165 (80.46)	359,497 (80.53)	
**Race (%)**					<0001
White	903 (76.44)	1803 (71.62)	316,785 (73.69)	319,491 (73.69)	
African American	165 (13.98)	444 (17.66)	62,518 (14.54)	63,127 (14.56)	
Hispanic	84 (7.13)	209 (8.31)	43,438 (10.1)	43,732 (10.09)	
Asian or Pacific Islander	14 (1.19)	25 (0.99)	4910 (1.14)	4949 (1.14)	
Native American	15 (1.27)	36 (1.42)	2239 (0.52)	2290 (0.53)	
**Characteristics of Patients**					
**Median Household Income Category for patient’s Zip code (%) ***					0.0035
0–25th percentile	314 (25.94)	628 (24.65)	112,302 (25.96)	113,244 (25.95)	
26–50th percentile	277 (22.86)	634 (24.89)	108,189 (25.01)	109,100 (25)	
51–75th percentile	329 (27.12)	734 (28.8)	110,482 (25.54)	111,545 (25.56)	
76–100th percentile	292 (24.08)	552 (21.66)	101,667 (23.5)	102,510 (23.49)	
**Primary Payer (%)**					<0.0001
Medicare	239 (19.54)	637 (24.8)	79,516 (17.99)	80,392 (18.04)	
Medicaid	245 (20.03)	350 (13.63)	66,603 (15.07)	67,198 (15.08)	
Private Insurance	598 (48.79)	1377 (53.65)	240,928 (54.52)	242,904 (54.49)	
Other/Self-pay/No charge	143 (11.64)	203 (7.92)	54,896 (12.42)	55,242 (12.39)	
**Admission type (%)**					0.0002
Non-elective	1148 (93.66)	2301 (89.98)	397,980 (90.14)	401,429 (90.15)	
Elective	78 (6.34)	256 (10.02)	43,527 (9.86)	43,861 (9.85)	
**Admission day (%)**					<0.0001
Weekday	894 (72.97)	1993 (77.62)	351,404 (79.39)	354,291 (79.36)	
Weekend	331 (27.03)	575 (22.38)	91,248 (20.61)	92,154 (20.64)	
**Characteristics of Hospitals**					
**Bedsize of hospital (%) ^†^**					<0.0001
Small	202 (16.45)	208 (8.18)	49,365 (11.21)	49,775 (11.2)	
Medium	298 (24.28)	646 (25.39)	108,130 (24.55)	109,073 (24.55)	
Large	726 (59.27)	1690 (66.43)	282,980 (64.24)	285,397 (64.24)	
**Hospital Location and Teaching Status (%)**					<0.0001
Rural	105 (8.58)	156 (6.12)	39,038 (8.86)	39,299 (8.85)	
Urban Non-teaching	542 (44.23)	893 (35.12)	182,840 (41.51)	184,275 (41.48)	
Urban Teaching	578 (47.19)	1495 (58.76)	218,597 (49.63)	220,670 (49.67)	
**Hospital Region (%)**					<0.0001
Northeast	168 (13.73)	409 (15.94)	100,147 (22.62)	100,724 (22.56)	
Midwest	240 (19.6)	849 (33.06)	84,468 (19.08)	85,557 (19.16)	
South	560 (45.64)	996 (38.81)	192,476 (43.48)	194,032 (43.46)	
West	258 (21.03)	313 (12.19)	65,561 (14.81)	66,132 (14.81)	
**Comorbidities of Patients (%)**					
Diabetes	130 (10.57)	467 (18.19)	56,438 (12.84)	57,034 (12.86)	<0.0001
Drug abuse	84 (6.87)	118 (4.59)	18,446 (4.2)	18,648 (4.2)	<0.0001
Obesity	198 (16.14)	526 (20.48)	47,065 (10.7)	47,789 (10.78)	<0.0001
Hypertension	342 (27.94)	1172 (45.67)	160,282 (36.45)	161,796 (36.48)	<0.0001
Renal failure	61 (4.96)	175 (6.83)	8447 (1.92)	8683 (1.960	<0.0001
**Deyo’s Charlson Comorbidity Index (CCI)**					<0.0001
0	807 (65.8)	1341 (52.23)	290,765 (65.69)	292,912 (65.61)	
1	262 (21.37)	625 (24.36)	98,398 (22.23)	99,285 (22.24)	
2	88 (7.15)	341 (13.29)	33,718 (7.62)	34,147 (7.65)	
3	40 (3.23)	144 (5.63)	11,203 (2.53)	11,387 (2.55)	
4	≤10	80 (3.12)	3821 (0.86)	3912 (0.88)	
≥5	20 (1.61)	35 (1.37)	4747 (1.07)	4801 (1.08)	

* This represents a quartile classification of the estimated median household income of residents in the patient’s ZIP Code; ^†^ Bedsize of hospital indicates number of hospital beds which varies depends on hospital location (Rural/Urban), teaching status (Teaching/Non-teaching) and Region (Northeast/Midwest/Southern/Western); percentage in brackets are column; percentage indicates direct comparison between Deficiencies vs. No-Deficiencies amongst patients with migraine from year 2003–2014.

**Table 2 medicina-55-00407-t002:** Univariate analysis of outcomes hypocalcemia and vitamin D deficient patients.

	Hypocalcemia	Vitamin D Deficiency	No Deficiency	Total	*p* Value
**APRDRG Severity or Disability/Loss of Function (%)**					<0.0001
Minor loss of function	0 (0)	0 (0)	219,797 (50)	219,797 (49.58)	
Moderate loss of function	891 (72.65)	2031 (79.1)	191,531 (43.57)	194,452 (43.86)	
Major loss of function	330 (26.94)	506 (19.69)	27,231 (6.2)	28,067 (6.33)	
Severe loss of function	<10	31 (1.21)	1006 (0.23)	1042 (0.23)	
Total Major/Severe Loss of Function (%)	335 (27.35)	537 (20.9)	28,237 (6.43)	29,109 (6.56)	<0.0001
**Length of Stay ± SE (Days)**	3.6 ± 0.2	3.4 ± 0.1	2.8 ± 0.01		<0.001
**Cost of Hospitalization ± SE ($)**	22590 ± 13	24914 ± 10	18,185 ± 55		<0.001

APRDRG, All Patients Refined Diagnosis Related Groups; SNF, Skilled Nursing Facility; ICF, Intermediate Care Facility; SE, standard error; percentage in brackets are column; percentage indicates direct comparison between Deficiencies vs. No-Deficiencies amongst patients with migraine from year 2003–2014.

**Table 3 medicina-55-00407-t003:** Logistic regression analysis showing linkage between disability and deficiencies.

**A: Un-Adjusted Multivariable Logistic Regression Analysis Showing Linkage Between Disability and Deficiencies.**					
	**Odds Ratio (OR)**	**Confidence Interval (CI)**		***p* Value**	**Area under the ROC Curve/c-Index**
		**LL**	**UL**		
**No Deficiency**	Reference				-
**Model 1: Disability due to Vitamin D Deficiency**	3.79	3.10	4.70	<0.0001	**0.507**
**Model 2: Disability due to Hypocalcemia**	5.53	4.14	7.37	<0.0001	**0.505**
**B: Adjusted Multivariable Logistic Regression Analysis to Predict Disability amongst Migraine Patients. ***					
	**Odds Ratio (OR)**	**Confidence Interval (CI)**			***p* Value**
		**LL**	**UL**		
**No Deficiency**	Reference				
**Vitamin D Deficiency**	3.12	2.38	4.08		<0.0001
**Hypocalcemia**	6.19	4.40	8.70		<0.0001
**Age (Every 10-Years)**	0.99	0.99	1.00		0.0492
**Gender**					
Female	Reference				
Male	1.04	0.96	1.13		0.3466
**Race**					
White	Reference				
African American	0.96	0.87	1.05		0.2047
Hispanic	0.71	0.63	0.80		<0.0001
Asian or Pacific Islander	0.65	0.47	0.89		0.001
Native American	0.67	0.43	1.06		0.0491
**Median Household Income Category for patient’s Zip code**					
0–25th percentile	Reference				
26–50th percentile	1.11	1.01	1.21		0.0181
51–75th percentile	1.13	1.03	1.23		0.0109
76–100th percentile	1.12	1.02	1.23		0.0209
**Primary Payer**					
Medicare	Reference				
Medicaid	0.77	0.69	0.86		<0.0001
Private Insurance	0.65	0.59	0.71		<0.0001
Other/Self-pay/No charge	0.60	0.53	0.68		<0.0001
**Admission type**					
Non-elective	Reference				
Elective	0.78	0.69	0.88		<0.0001
**Admission day**					
Weekday	Reference				
Weekend	1.12	1.04	1.21		0.003
**Bedsize of hospital**					
Small	Reference				
Medium	0.92	0.82	1.03		0.1383
Large	1.04	0.94	1.15		0.5121
**Hospital Location and Teaching Status**					
Rural	Reference				
Urban Non-teaching	0.97	0.85	1.10		0.6378
Urban Teaching	1.17	1.03	1.33		0.013
**Hospital Region**					
Northeast	Reference				
Midwest	1.21	1.09	1.35		0.0003
South	1.21	1.10	1.32		<0.0001
West	1.49	1.34	1.66		<0.0001
**Comorbidities of Patients**					
Drug abuse	2.02	0.94	4.30		0.1065
Obesity	2.47	1.64	3.73		0.0701
Hypertension	3.16	2.42	4.11		<0.0001
Renal failure	2.53	2.15	2.97		<0.0001
Diabetes	6.59	0.67	65.12		<0.0001
Alcohol Abuse/dependence	1.38	1.07	1.77		0.0132
Smoking status	0.96	0.88	1.04		0.2723
**Deyo’s Charlson Comorbidity Index (CCI)**					
1	Reference				
0	0.43	0.39	0.46		<0.0001
2	1.95	1.76	2.15		<0.0001
3	3.29	2.87	3.78		<0.0001
4	5.60	4.54	6.90		<0.0001
≥5	5.42	4.33	6.78		<0.0001
**Area under the ROC curve/c-index**	**0.819**				

OR, Odds Ratio; CI, Confidence Interval; UL, Upper Limit; LL, Lower Limit, Model in Table 3A is an unadjusted model. * Final Model (Table 3B) is adjusted for basic demographic with patient-level variables, comorbidities, CCI, concurrent conditions, and hospital-level variables such as hospital region, teaching status, and bed size.

**Table 4 medicina-55-00407-t004:** Analysis showing linkage between deficiencies and migraine in 2014 hospitalizations.

**A: Univariate Analysis of Linkage Between Deficiencies and Migraine.**						
**Frequency (%)**	**Hypocalcemia**	**Vitamin D Deficiency**	**No Deficiency**	**Total**		***p* Value**
**No-Migraine**	238,885 (98.5)	406,400 (97)	27,105,780 (98.4)	27,751,065		<0.0001
**Migraine**	3630 (1.5)	12,570 (3)	445,555 (1.6)	461,755		
	242,515	418,970	27,551,335	28,212,820		
**B: Adjusted Multivariable Logistic Regression Analysis to Predict Linkage Between Deficiencies and Migraine**						
		**Odds Ratio (OR)**		**Confidence Interval (CI)**		***p* Value**
				**LL**	**UL**	
**No Deficiency**		Reference				
**Vitamin D Deficiency**		1.97		1.89	2.05	<0.0001
**Hypocalcemia**		1.11		1.03	1.20	0.0061
**Age (Every 10-Years)**		0.98		0.98	0.98	<0.0001
**Gender**						
Female		Reference				
Male		0.31		0.31	0.32	<0.0001
**Race**						
White		Reference				
African American		0.64		0.62	0.65	<0.0001
Hispanic		0.60		0.58	0.62	<0.0001
Asian or Pacific Islander		0.37		0.34	0.39	<0.0001
Native American		0.85		0.78	0.93	0.0002
**Median Household Income Category for patient’s Zip code**						
0–25th percentile		Reference				
26–50th percentile		1.09		1.07	1.11	<0.0001
51–75th percentile		1.11		1.09	1.13	<0.0001
76–100th percentile		1.18		1.16	1.21	<0.0001
**Primary Payer**						
Medicare		Reference				
Medicaid		0.90		0.88	0.93	<0.0001
Private Insurance		1.30		1.27	1.33	<0.0001
Other/Self-pay/No charge		1.08		1.04	1.11	<0.0001
**Admission type**						
Non-elective		Reference				
Elective		0.81		0.80	0.83	<0.0001
**Admission day**						
Weekday		Reference				
Weekend		0.92		0.91	0.94	<0.0001
**Bed size of hospital**						
Small		Reference				
Medium		0.99		0.97	1.01	0.1675
Large		1.06		1.04	1.08	<0.0001
**Hospital Location and Teaching Status**						
Rural		Reference				
Urban Non-teaching		1.24		1.21	1.28	<0.0001
Urban Teaching		1.43		1.40	1.47	<0.0001
**Hospital Region**						
Northeast		Reference				
Midwest		1.03		1.01	1.05	0.0056
South		1.02		1.00	1.04	0.0617
West		1.08		1.05	1.10	<0.0001
**Comorbidities of Patients**						
Drug abuse		1.54		1.50	1.58	<0.0001
Obesity		1.42		1.40	1.45	<0.0001
Hypertension		1.35		1.32	1.38	<0.0001
Renal failure		0.69		0.67	0.71	<0.0001
Diabetes		0.68		0.67	0.70	<0.0001
Hypercholesterolemia/lipidemia		1.33		1.29	1.37	<0.0001
Alcohol Abuse/dependence		0.73		0.70	0.75	<0.0001
Current or Past Smoker		1.33		1.31	1.35	<0.0001
**Deyo’s Charlson Comorbidity Index (CCI)**						
**1**		Reference				
0		0.56		0.55	0.58	<0.0001
2		0.90		0.88	0.92	<0.0001
3		0.83		0.81	0.86	<0.0001
4		0.68		0.65	0.71	<0.0001
≥5		0.55		0.53	0.58	<0.0001
**Area under the ROC curve/c-index**		**0.725**				

OR, Odds Ratio; CI, Confidence Interval; UL, Upper Limit; LL, Lower Limit.

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
