# Peer review of "Hypocalcemia and Vitamin D Deficiency amongst Migraine Patients: A Nationwide Retrospective Study"

_medicina, 2019, doi:10.3390/medicina55080407_

Round 1

Reviewer 1 Report

This manuscript conducted by Urvish Patel and colleagues, they analyzed data from a large number of inpatients,

Revealed Vitamin D deficiency and hypocalcemia had potential connection with migraine. This article had comprehensive and persuasive data, also the author had a heavy workload. Overall, this manuscript is a very good job.

Highlight:

1 Huge medical record information.

2 The cost of migraineurs with Vitamin D deficiency and hypocalcemia were higher then the NO deficiency group.

3 Compared with a large number of groups,such as gender, age, race and different degrees of Vitamin D dysfunction. The analysis is meticulous and reliable.

The deficiencies are as follows:

1 The data in the paper are concentrated from 2003 to 2014, but there is no data in the last five years. Although the author also mentioned this matter, it is still insufficient.

2 Less than one sixth of the references are published in the last five years.It is recommended to increase the number of recent references to make the article more scholarly.

3 Some minor typographical errors. Line60 19346 should be 1934 [6].

Author Response

I appreciate the time and efforts you have given to review the manuscript. We have reviewed the comments.

1 The data in the paper are concentrated from 2003 to 2014, but there is no data in the last five years. Although the author also mentioned this matter, it is still insufficient.

- Though USA Nationwide Inpatient Sample available to use up to the year 2016. Accuracy of ICD 10 codes utilized in 2015 and 2016 are not known. So we are working on it. We would not like to submit the result of the years where accuracy is unknown. 

https://www.distributor.hcup-us.ahrq.gov/Databases.aspx

2 Less than one-sixth of the references are published in the last five years. It is recommended to increase the number of recent references to make the article more scholarly.

- In the next draft, we will try to replace all the references before the year 2003. As the population of interest starting from 2003, It would be interesting to compare the literature starting from 2003 onwards.

3 Some minor typographical errors. Line60 19346 should be 1934 [6].

- Next draft, we have fixed it.

Reviewer 2 Report

The Authors present a well-documented paper providing clinical data from hospitalized migraineurs and patients with hypocalcemia and vitamin D deficiency. The authors draw some conclusions linking hypocalcemia and vitamin D to the severity of migraines and hospitalization. Overall, the paper serves as an exploratory look at the data, to provide some evidence for further clinical trials (as aptly mentioned by the author). Taken with the known pitfalls, the paper is well presented as an exploratory document. I only recommend the title be reworded to better reflect the absolute conclusions of the paper, as currently is misleading and a bit exaggerated.        

Author Response

I appreciate your effort to review the manuscript. here is your comment-

"I only recommend the title be reworded to better reflect the absolute conclusions of the paper, as currently is misleading and a bit exaggerated."

We are open to accepting suggestions if you have any, We can switch to a more straight forward title "Hypocalcemia and Vitamin D deficiency are responsible with Migraine attacks and increase the disability amongst Migraineurs: A nationwide retrospective study."

Please let us know.

Reviewer 3 Report

The debate among the existing relationship between vitamin D and calcium intake in chronic headaches goes on from a long time. Surely nutrition habits can influence as external trigger factor the onset of migraine crisis. This study has the merit to confirm that migraine patients (a wide cohort of 461.755) present problems related to vitamin C deficiency and hypocalcemia. The study on big populations can give interesting results but it can also underestimate co-occurencies. 

It is important to report also that in cluster headache also there are similar dficencies (PMID: 30019090)

Author Response

We appreciate your time to review the article and give your valuable feedback.

It is important to report also that in cluster headache also there are similar deficiencies (PMID: 30019090) feedback.

- We have added this article in the next version.